# Study on air temperature estimation and its influencing factors in a complex mountainous area

**Wang Runke** *, **You Xiaoni, Shi Yaya, Wu Chengyong, Liu Baokang**

College of Resources and Environmental Engineering, Tianshui Normal University, Tianshui, Gansu, China

* wrk0531@tsnu.edu.cn

## Abstract

Near-surface air temperature (Ta) is an important parameter in agricultural production and climate change. Satellite remote sensing data provide an effective way to estimate regional-scale air temperature. Therefore, taking Gansu section of the upper Weihe River Basin as the study area, using the filtered reconstructed high-quality long-time series normalized difference vegetation index (NDVI), interpolated reconstructed land surface temperature (LST), surface albedo, and digital elevation model (DEM) as the input data, the back-propagation artificial neural network algorithm (BP-ANN) was combined with a multiple linear regression method to estimate regional air temperature, and the influencing factors of air temperature estimation were analyzed. This method effectively compensates for the fact that air temperature data provided by a single station cannot represent regional air temperature information. The result shows that the temperature estimation accuracy is high. In terms of interannual variation, the air temperature in the study area showed a slightly increasing trend, with an average annual increase of 0.047˚C. The calculation results of the interannual variation rate of temperature showed that the area with increased air temperature accounted for 75.8% of the total area. In terms of seasonal variation, compared with that in summer and winter, the air temperature rising trend in autumn was obvious, and the air temperature in the middle of the study area decreased in spring, which is prone to frost disasters. LST and NDVI in the study area were positively correlated with air temperature, and their positive correlation distribution areas accounted for 93.62% and 94.34% of the total study area, respectively. NDVI, LST and DEM influence the temperature change in the study area. The results show that there is a significant positive correlation between NDVI and air temperature, and the change of NDVI has a positive effect on the spatiotemporal variation of air temperature. The correlation coefficient between LST and air temperature in the southeast of the study area is negative, and there is a difference. In addition, the correlation coefficient between LST and air temperature in other areas of the study area is positive. The air temperature decreased with elevation, air temperature decreases by 0.27˚C every hundred meters.

**Data Availability Statement:** MODIS data were retrieved from NASA LAADS Web (https://ladsweb. modaps.eosdis.nasa.gov/search/). DEM data was provided by the Geospatial data cloud (https:// www.gscloud.cn/) Meteorological data are

obtained through China Meteorological Data
Service Centre (http://data.cma.cn/).

**Funding:** This research was supported by the
Science and Technology Program of Gansu
Province (21JR1RE293) and the School-Level
Scientific Research Project of Tianshui Normal
University (ZDY2020-18). The funders had no role
in study design, data collection and analysis,
decision to publish, or preparation of the
manuscript.

**Competing interests:** The authors have declared
that no competing interests exist.

## Introduction

The near-surface air temperature (Ta) is a key parameter for energy and water exchange
between the surface and the atmosphere [1]. The spatiotemporal distribution of air tempera-
ture provides valuable information for understanding and simulating complex surface pro-
cesses and revealing the possible changes in surface water, heat balance, and vegetation cover
that may be caused by climate change or local disturbance [2]. Although meteorological sta-
tions can provide long-term air temperature data, they are spatially dispersed, so the air tem-
perature data are only effective in a certain area near the station [3]. The lack of air
temperature data on a spatiotemporal scale limits the ability to analyze spatiotemporal changes
in air temperature in heterogeneous areas [4, 5]. Therefore, accurate estimation of spatiotem-
poral distributions of the air temperature is important.

For decades, several methods have been used to estimate air temperature at regional scales.
These methods can be divided into three categories: (1) interpolation, (2) land surface energy
balance, and (3) statistical methods. The methods of air temperature interpolation includes the
inverse distance weight method (IDW), spline interpolation method, kriging, and other meth-
ods, among which the ordinary kriging method is better than other methods [6, 7]. The advan-
tage of the interpolation method is fast but is limited by the density of stations and the
complexity of different environmental conditions [8]; the more the data, the more accurate the
interpolation result, and the fewer the data, the larger the error. In addition, the interpolation
of meteorological stations is carried out in a closed environment, including stations, and local
nonstationarity and thus deviation may be introduced in the interpolation process [9]. The
land surface energy balance method [10] is one of the methods used to estimate near-surface
air temperature based on land surface temperature (LST). According to the land energy bal-
ance equation, LST and other surface environmental parameters such as emissivity, albedo,
and surface heat flux are combined to estimate the air temperature [11, 12]. The advantage of
this method is that it provides a solid theoretical basis. Zakšek et al. [13] showed that this
method can estimate air temperature with high spatiotemporal resolution at a height of 2 m,
which is an effective method. The disadvantage of this method is that it is not universal, and
the parameters must be re-fitted when applied to different regions. When the statistical
method is used to estimate air temperature on a regional scale, LST is the most relevant remote
sensing data for estimating air temperature [14–17]. The statistical method makes it possible
to explain spatial differences in air temperature. It can provide a feasible method to improve
the temporal and spatial accuracy of air temperature estimation, which is attributed to its spa-
tial continuity [18]. Statistical methods include single-factor [8] and multi-factor statistical
methods [19, 20]. The single-factor method is relatively simple. The statistical model provides
only a linear regression between LST and air temperature [21], but the estimation accuracy is
not high. Compared with that of the single-factor method, the calculation of the multi-factor
statistical method is complex and requires stepwise regression between variables to build a sta-
tistical model to estimate the air temperature [22], which effectively improves the accuracy of
air temperature simulation [23]. For example, a multiple linear regression model was estab-
lished using MODIS LST products to estimate the maximum, minimum, and average air tem-
perature in northeast China, and the predicted results were good [14]. Through a comparison
of the three methods, the multi-factor statistical method was found to be universal and robust
for air temperature estimation on a regional scale.

The normalized difference vegetation index (NDVI) is related to air temperature changes
[24]; however, in previous studies on air temperature estimation, NDVI data under clear sky
conditions were selected, and the missing and abnormal data of NDVI pixel values were dis-
carded [25]. However, remote-sensing data are generally affected by clouds, rain, and snow.

To ensure the continuity of NDVI data in time and space, it is necessary to deal with abnormal NDVI data, that is, to reconstruct NDVI time series data. Research shows that data reconstruction can effectively improve the accuracy of air temperature estimation results when NDVI data are used as input data [25]. Savitzky Golay (S-G) smoothing filtering [26] is an important method for realizing NDVI data reconstruction. The S-G filtering method has obvious advantages and can process NDVI data in the time domain. The reconstructed NDVI data result is better [27], as it is closer to the upper envelope of the NDVI sequence [28]. After reconstructing the NDVI, high-quality NDVI time series data can be obtained, which can solve the discontinuity of vegetation coverage in space and time and improve the application of vegetation coverage in surface process simulation and ecosystem modeling.

The relationship between LST and air temperature is complex from the standpoint of theory and experience [29]; however, there is a strong correlation between them [30]. Currently, the most commonly used LST data are the MODIS LST products. Owing to the high accuracy of MODIS LST products, with an average accuracy of 1 K under clear sky conditions [31], they are widely used in surface process research [16, 32] However, affected by clouds and other atmospheric conditions, MODIS LST images contain invalid pixels with missing values [33]. Therefore, before using MODIS LST data, it is important to adopt effective methods to compensate for the defects caused by missing pixel values, solve the problem of obtaining LST information in cloud-covered areas, realize the integrity and continuity of LST data, and improve the effective utilization of remote sensing images. To eliminate the impact of data loss and poor data quality on data utilization, the commonly used LST interpolation and reconstruction methods include the neighboring pixel (NP) theoretical method [34], spatial interpolation method [35], and time series method estimation [36]. The shortcomings of the NP theoretical approach are the uncertainties generated by the errors in parameter estimation, net solar radiation inversion, and the inherent inaccuracy of the NP scheme. The spatial interpolation method only considers the values of the adjacent pixels. Nonetheless, their accuracy is significantly reduced when there is serious nearby data loss. The time series analysis method uses the characteristics of periodic changes in LST to reconstruct through filtering, but the LST data can be decomposed into harmonic variables of different periods only from the perspective of filtering, which will lose the mutation information of LST. However, LST is correlated with the digital elevation model (DEM) and NDVI [37]. Therefore, this study establishes a regression model with two factors, DEM and NDVI, related to LST and interpolates the missing pixels of LST to ensure the integrity of LST data.

Based on the abovementioned retrospective analysis, LST and NDVI affect the distribution and change in air temperature, and interpolation reconstruction of LST and NDVI is a prerequisite to ensure the spatiotemporal continuity of air temperature at the regional scale. This study has two main objectives: (1) to use NDVI and LST data obtained by data interpolation and reconstruction methods to realize the integrity and continuity of the data and (2) to use a back-propagation artificial neural network (BP-ANN) machine learning algorithm model, combined with a multiple linear regression method, NDVI, LST interpolation reconstruction data, and other remote sensing data to estimate air temperature and to analyze the factors influencing the air temperature estimation.

## Materials and methods

### Study area

The Gansu section of the upper Weihe River Basin is located at 34.18˚N–35.88˚N and 103.94˚E–106.74˚E, with an area of approximately $2.59 \times 10^4$ km$^2$ (Fig 1). The terrain of the study area is high in the northwest and low in the southeast. There were obvious regional

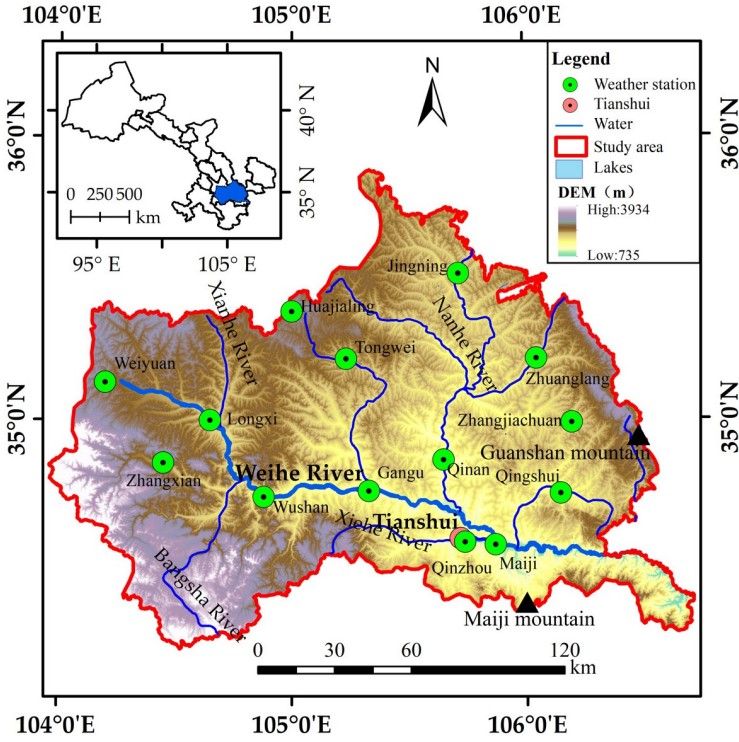

**Fig 1. Location of the study area.**

differences in geomorphology, including mountainous geomorphology in the east and south, loess hilly geomorphology in the north, Weihe River valley geomorphology in a small section of the central part due to the fault of the zonal tectonic belt, and the joint area of the eastern margin of the Gannan Plateau with the central Gansu Loess Plateau and Longnan Mountain in the west and southwest, with complex and diverse topography. The study area has a temperate monsoon climate, which is located in the transition zone from the semi-humid cold temperate zone to the semi-arid cold temperate zone, and is the intersection area of temperate continental, subtropical monsoon, and plateau alpine climates. The spatiotemporal distributions of hydrothermal conditions are uneven. For example, the hottest month in southeast Tianshui city is July, with an average air temperature of 22.8˚C. The coldest month is January, with an average air temperature of -2.0˚C. The annual average precipitation is 491.7 mm, with a substantial variance in precipitation between the southeast and the rest of the study region in space. Zhangxian County in the west has an average annual air temperature of 8.6˚C and precipitation of 458.3 mm. The Weihe River originates at Weiyuan, which is located in the northwest. The annual average air temperature is 6.8˚C, with maximum and minimum air temperatures of 30.5˚C and -20.1˚C, respectively, and an average precipitation of about 500 mm. The study area was a typical dry farming area with a fragile ecological environment consisting of 14 meteorological stations. In this study, the four seasons were defined as follows: spring from March to May, summer from June to August, autumn from September to November, and winter from December to February.

## Data

**Remote sensing data.** The remote sensing data in this study were MODIS data, including MOD11A2 (Version 6), MOD13A2, and MCD12Q1 data. MOD11A2 has a spatial resolution

of 1000 m and a temporal resolution of 8d. The MOD11A2 product dataset includes LST, quality control for LST and emissivity (QC), emissivity, and other data. The value of the LST pixel ranged from 7500 to 65535 (unit: K), and that of QC ranged from 0 to 255. A total of 958 scenes MOD11A2 data were collected for this study. MOD13A2 has a spatial resolution of 1000 m and temporal resolution of 16d, with 480 scenes. MCD12Q1 is a land use/land cover change product with a temporal resolution of 1a and a spatial resolution of 500 m. There were 17 land-cover types in total. All remote sensing data were projected and formatted through MODIS reprojection tools (MRT) software [38], and the reprojected coordinate system is "WGS 84/UTM Zone 48N." To make MODIS data consistent at the spatiotemporal scale, we interpolated and resampled MOD13A2 NDVI data to obtain 8d NDVI data. MCD12Q1 was resampled to 1 km for consistency with the spatial resolution of the other data.

DEM data were obtained from a geospatial data cloud with a spatial resolution of 90 m. After resampling the DEM to 1000 m, the DEM data required by the study were obtained.

**Ground data.** Ground meteorological data were obtained from 14 meteorological stations in the study area (Table 1). The observation data included air temperature, air pressure, and ground observation temperature at 0 cm, and the data collection interval was 1 hour. To keep the data compatible with the satellite's transit period, meteorological data were collected between 10:00 and 11:00 am, and the average value was calculated. Land surface air temperature observations data were divided into two categories. The data from 10 meteorological stations were used to drive the model that estimated the near-surface temperature, while the data from four meteorological stations in Qinan, Wushan, Qingshui and Weiyuan were used to verify the air temperature estimation results.

## Methods

**Air temperature estimation.** Air temperature is susceptible to the surrounding environment. Therefore, when estimating air temperature, we must consider the influence of multiple factors, such as ground features and environment [39], including LST, NDVI, albedo, and DEM. Based on this, the estimation of air temperature was divided into three steps: (1) NDVI long time series data filtering reconstruction, (2) interpolation of LST missing data, and (3) air temperature estimation using the BP-ANN machine learning algorithm. The overall technical procedure of this study is shown in Fig 2.

**Table 1. Meteorological stations and data information.**

| No, | Station | Longitude | Latitude | Elevation (m) | Date Range | Time (am) |
|---|---|---|---|---|---|---|
| 1 | Gangu | 105.33 | 34.75 | 1271.90 | 2006–2020 | 10:00 11:00 |
| 2 | Qinan | 105.65 | 34.86 | 1216.10 | 2006–2020 | 10:00 11:00 |
| 3 | Wushang | 104.88 | 34.73 | 1495.40 | 2006–2020 | 10:00 11:00 |
| 4 | Qinzhou | 105.74 | 34.57 | 1149.80 | 2006–2020 | 10:00 11:00 |
| 5 | Qingshui | 106.15 | 34.74 | 1416.10 | 2006–2020 | 10:00 11:00 |
| 6 | Zhangjiachuan | 106.2 | 34.99 | 1664.50 | 2006–2020 | 10:00 11:00 |
| 7 | Maiji | 105.87 | 34.56 | 1085.20 | 2006–2020 | 10:00 11:00 |
| 8 | Weiyuan | 104.2 | 35.13 | 2111.60 | 2015–2020 | 10:00 11:00 |
| 9 | Jingning | 105.71 | 35.51 | 1653.80 | 2015–2020 | 10:00 11:00 |
| 10 | Tongwei | 105.23 | 35.21 | 1768.90 | 2015–2020 | 10:00 11:00 |
| 11 | Zhuanglang | 106.0 | 35.21 | 1615.60 | 2015–2020 | 10:00 11:00 |
| 12 | Zhangxian | 104.45 | 34.85 | 1883.20 | 2015–2020 | 10:00 11:00 |
| 13 | Longxi | 104.65 | 35.00 | 1728.80 | 2015–2020 | 10:00 11:00 |
| 14 | Huajialing | 105 | 35.38 | 2450.40 | 2015–2016 | 10:00 11:00 |

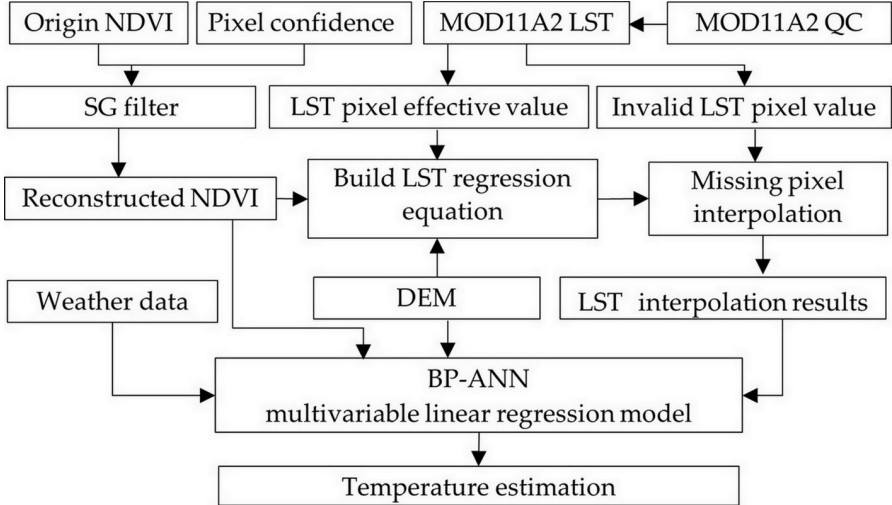

**Fig 2. The flow chart of temperature estimation.**

The BP-ANN algorithm was used for air temperature estimation. BP-ANN is the most representative and widely used artificial neural network at present. Typically, a BP neural network includes an input layer, output layer, and hidden layer [40]. The essence of the BP neural network algorithm is to determine a set of weights to minimize the error between the pre-rule value and the real value of the training set samples. Its multidimensional global optimization strategy improves not only the inversion speed of surface parameters but also the inversion accuracy to a certain extent. The specific calculation process is based on the BP-ANN algorithm, which selects the air temperature data of the meteorological station, the interpolated reconstructed LST and NDVI data of the corresponding station, and albedo and DEM as the input data, in which 80% of the data are selected for training and 20% of the data are used for testing, constructs a linear regression relationship between Ta and LST, NDVI, albedo, and DEM, and finally, this expression is used to estimate the temperature. The temperature estimation formula is as follows:

$$T_a = aLST + bNDVI + cAlbedo + dDEM + e \tag{1}$$

where Ta is the air temperature at the satellite passing time (˚C), *a, b, c,* and *d* is regression coefficients, *e* is a constant term, and the unit of DEM is km.

*NDVI time series data reconstruction.* MOD13A2 provides NDVI data (Band 1) and quality control data (Band 12). According to the quality data, the reconstruction of NDVI can be realized using certain mathematical methods. S-G smoothing filtering [26] is an important method for realizing NDVI data reconstruction. The basic principle of S-G filtering is to select five points, including $X_m$ as the $X$ set. Polynomial smoothing filtering uses the data of points $X_m - 2$, $X_m - 1$, $X_m$, $X_m + 1$, and $X_m + 2$ to fit a polynomial, replace $X_m$ with the polynomial fitting value and move in turn until the data are traversed. When S-G approximates the basic function, it does not use a constant window but uses a higher-order polynomial to realize polynomial least squares fitting in the sliding window. The polynomial is designed to preserve higher moments in the data and reduce the bias introduced by the filter [41].

Based on the S-G filtering principle, the reconstruction formula for the NDVI time-series data is as follows:

$$NDVI_{f_j} = \frac{1}{N} \sum_{i=-m}^{+m} NDVI_{j+i} C_i \tag{2}$$

where $NDVI_{j+i}$ is the original NDVI value, $NDVI_{f_j}$ is the reconstructed NDVI value, $C_i$ is the coefficient of the $i^{th}$ original NDVI value of the filter (smoothing window), $j$ is the index of the original NDVI data list, and $n$ is the number of convolution integers, equal to the size of the smoothing window ($2m+1$). The filter size consists of $2m+1$ points, where $m$ is the half width of the smoothing window. The key to S-G convolution smoothing lies in the solution of the matrix operator.

*Spatial Interpolation of MODIS LST Products*. The interpolation of the MODIS LST missing pixels was realized using moving window regression interpolation. The specific implementation process involves setting the missing pixels, which are composed of two parts. One part is to take the QC image as the mask data, find the pixel corresponding to QC = 2 (the pixel covered by cloud interference or cloud) on the LST image and set it as a null value as the missing pixel. The other part is to find the value of LST pixel in the LST image that is not within the range of effective value (less than 7500 and greater than 65535), These low-quality pixel values are set as null values as missing pixels, and high-quality LST pixels are retained after processing. Taking the missing pixel as the center and a certain distance as the radius, the effective surface temperature within the radius (the pixels within the radius constitute a window) was searched through the quality control data, and the multiple regression relationship between the effective values of LST, NDVI, and DEM was constructed for interpolation. After the interpolation of the current missing pixel is completed, we move to the next missing pixel and repeat the preceding steps for interpolation, while the NDVI and DEM corresponding to the missing pixel of the LST value exist. Therefore, the relationship created by the effective LST can be applied to the missing data to obtain the LST of the missing pixels. The multiple regression expression constructed using the LST effective value is:

$$LST = a * NDVI + b * DEM + c \tag{3}$$

where $a$ and $b$ are regression coefficients and $c$ is a constant term. In the moving interpolation window, the values of $a$, $b$, and $c$ change with each window move. Simultaneously, land use type data (MCD12Q1) were used in the interpolation process, and the influence of the water body on LST was considered.

*Evaluation of air temperature estimation model*. Limited by the number of ground observation stations, in order to evaluate the performance of the air temperature estimation model constructed by BP-ANN method, k-fold cross validation method is used for validation [42]. The k-fold cross validation is a popular method to evaluate the performance of algorithms [43], and the value of k is usually ten. The ten-fold cross validation will perform a total of 10 fitting procedures. Each fitting will be performed on the training set randomly selected by 90% of the total training set, and the remaining 10% will be used as the set to be verified [44].

**The evaluation index.** *Coefficient of variation*. The variation in NDVI has temporal and spatial variabilities. On a spatial scale, the variation in NDVI in different regions was not consistent. In this study, the coefficient of variation (CV) was used to measure the difference in spatial variation, which determines the dispersion or variability of the data [45], using the

following formula [46]:

$$CV = \sigma/\mu \tag{4}$$

where $\sigma$ is the standard deviation, and $\mu$ is the average value.

*Accuracy evaluation index.* The difference between the calculated and observed values can be evaluated using the root mean square error (RMSE), mean absolute error (MAE) [47] and correlation coefficient (*R*). The smaller value of RMSE, the more accurate the calculation result of the model. The smaller value of the MAE, the smaller the error between the calculated value and the observed value. The closer the correlation coefficient *R* is to 1.0, the higher the calculation accuracy of the model.

**Analysis methods.** *Spatiotemporal scale analysis method.* To quantify the change patterns of LST and Ta in time and space, the least-squares linear regression method was used to calculate the slope of the unary linear equation of LST and Ta from 2001 to 2020. The slope is the long-term trend, *k*. When *k* >0, an upward trend is indicated. When *k* <0, it exhibited a downward trend. The calculation formula is [48]

$$k = \frac{\sum_{u=1}^{m} (uT_u) - \sum_{u=1}^{m} u \sum_{u=1}^{m} T_u/m}{\sum_{u=1}^{m} u^2 - \left(\sum_{u=1}^{m} u\right)^2/m} \tag{5}$$

where *u* is the year serial number, $T_u$ is the value of LST or Ta corresponding to the year *u*, *m* is the research period, and *m* = 20.

*Mann–Kendall Test.* The Mann–Kendall (M-K) test statistic *S* is given as follows [49, 50]:

$$S = \sum_{i=1}^{n-1} \sum_{j=i+1}^{n} sign\left(x_j - x_i\right) \tag{6}$$

where *n* is the length of the dataset, and $x_i$ and $x_j$ are the data values in the time series *i* and *j* (*j* > *i*), respectively, where the sign is a sign function. The term "sign" is defined as

$$sign\left(x_j - x_i\right) = \begin{cases} +1 & x_j - x_i > 0 \\ 0 & x_j - x_i = 0 \\ -1 & x_j - x_i < 0 \end{cases} \tag{7}$$

## Results

### Air temperature estimation results and verification

In this study, based on the BP-ANN algorithm, observed data from 9580 meteorological stations were selected to establish the equation, and 3832 observed data points were verified. In this study, based on the BP-ANN algorithm, 9580 observation data obtained from meteorological stations were selected to establish the equation, and 3832 observed data points were verified. A stepwise regression analysis was adopted to obtain the temperature estimation formula, expressed as

$$T_a = 0.65LST + 19.23NDVI + 0.17\alpha - 2.84h - 3.81 \tag{8}$$

where LST is the land surface temperature (˚C), NDVI is the normalized difference vegetation index, $\alpha$ is the surface albedo, and *h* is elevation (km).

The air temperature data verified by the results were obtained from the meteorological stations in Qinan, Wushan, Qingshui, and Weiyuan. The same observation data as the satellite transit time were selected, and the comparison between the two is based on the geographical position (latitude and longitude) point to point. The verification results showed that the accuracy between the observed and estimated values were accurate. RMSE values of Qinan,

Wushan, Qingshui and Weiyuan stations were 2.33˚C, 2.28˚C, 3.02˚C, and 1.86˚C, respectively (Fig 3). Except for Qingshui station, the RMSE of the other three stations was less than 2.5˚C. The MAE values of the four stations were 1.8, 1.79, 1.51, and 1.43˚C, respectively. MAE is less than 2˚C, and MAE is less than RMSE, which indicates that the error of temperature estimation is small and the estimation results are consistent. The observed and estimated values of the Qinan and Wushan stations were close to both sides of the 1:1 line, but when the air temperature was greater than 10˚C, the estimated values of the Qinan and Wushan stations were slightly lower than the observed values (Fig 3A and 3B). The scatter plot of the Qingshui station shows that most of the points are above the 1:1 line, and the estimated air temperature is slightly higher than the observed value (Fig 3C). The scatter diagram of the Weiyuan station is close to both sides of the 1:1 line, and the RMSE and MAE are both less than 2˚C. The air temperature estimation accuracy of this station was relatively high (Fig 3D). Overall, compared with the estimated values, the observed values have a good correlation, and the correlation coefficients, *R*, are 0.97, 0.98, 0.97, and 0.98, respectively. This shows that the temperature estimation results were good and reasonable.

The model was established by using the method of ten-fold cross validation, and the estimation formula of air temperature was obtained, which was expressed as

$$T_a = 0.78LST + 1.82NDVI - 0.024\alpha - 1.83h - 2.21 \qquad (9)$$

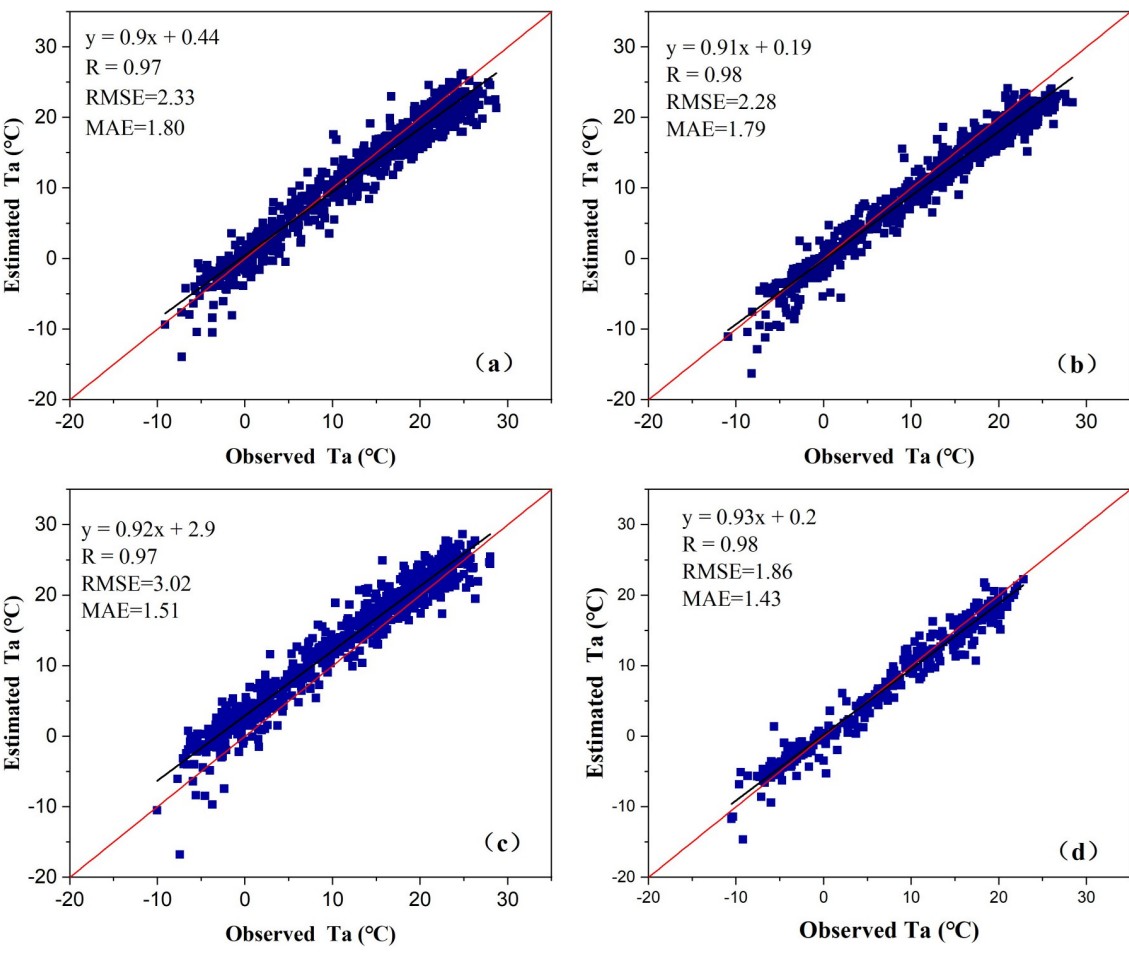

**Fig 3. Comparison between observed and calculated temperature values.** (a) Qinan, (b) Wushan, (c) Qingshui, and (d) Weiyuan.

where LST is the land surface temperature (°C), NDVI is the normalized difference vegetation index, α is the surface albedo, and *h* is elevation (km).

The Fig 4 shows the comparison between the calculated and observed air temperature values obtained by the ten-fold cross validation method. The RMSE values of Qinan, Wushan, Qingshui and Weiyuan stations were 3.58°C, 2.78°C, 3.51°C, and 3.09°C, The MAE values of the four stations were 2.65, 2.14, 3.28, and 3.09°C, respectively. The RMSE and MAE values of ten-fold cross validation are greater than those of BP-ANN. the correlation coefficients, *R*, are 0.94, 0.94, 0.92, and 0.94, respectively, this indicates that the accuracy of the ten-fold cross validation method is slightly lower than that of the BP-ANN method. On the whole, the BP-ANN algorithm is better than the ten-fold cross validation method in air temperature estimation, and the air temperature estimation model fitted by BP-ANN method has better performance and higher accuracy.

## Temporal and spatial variation characteristics of air temperature

Fig 5 shows the multi-year average value and long-term trend of air temperature in each pixel calculated using the air temperature estimation results from 2001 to 2020. In terms of spatial distribution, the overall trend of the multi-year average air temperature distribution showed a

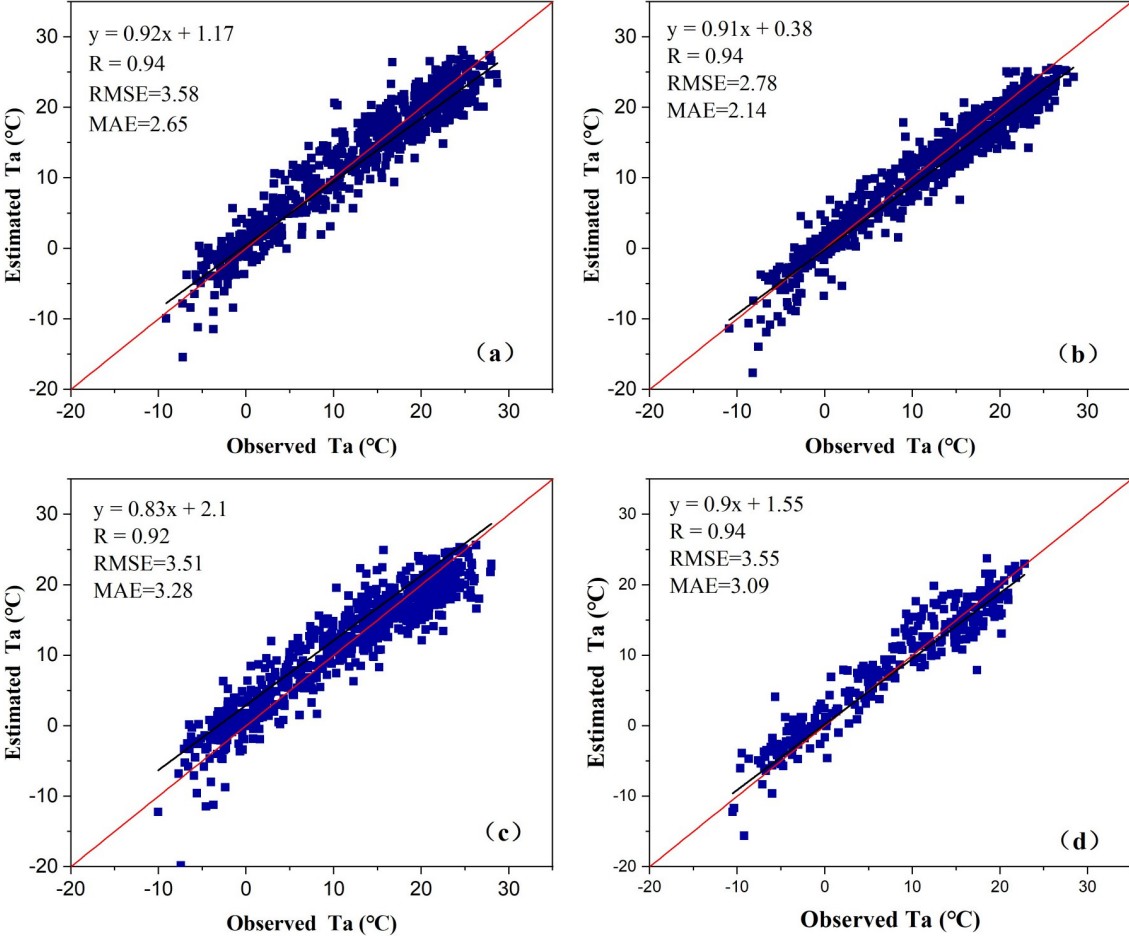

**Fig 4. Comparison between the air temperature calculated values obtained by ten-fold cross validation method and observed values.** (a) Qinan, (b) Wushan, (c) Qingshui, and (d) Weiyuan.

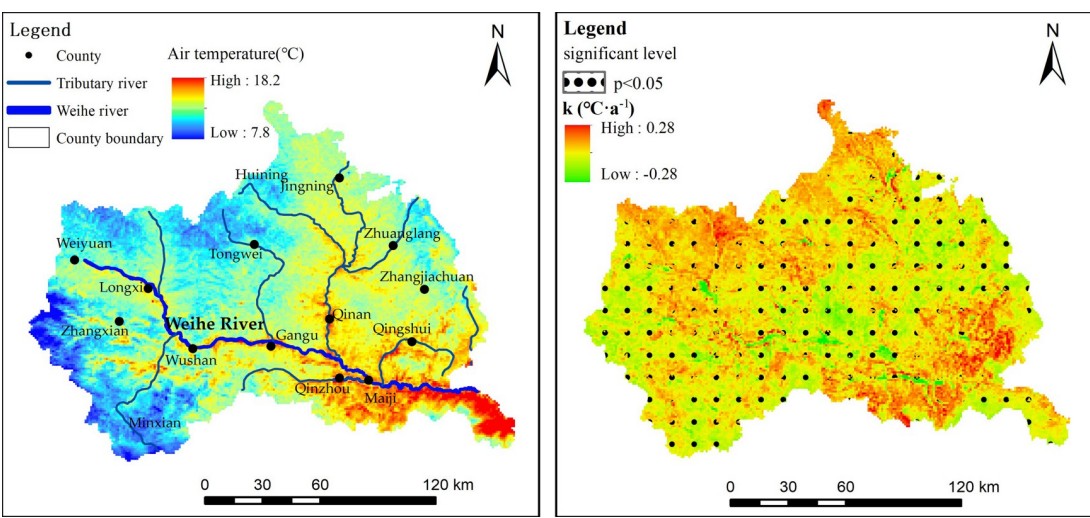

**Fig 5. Spatial distribution of average temperature and long-term trend from 2001 to 2020.**

decreasing trend from southeast to northwest and from east to west (Fig 5A). Higher average air temperatures are found where rivers flow, such as in the river valley where the Weihe River and its tributaries flow. The highest average air temperature, 18.2˚C, was recorded in the southeast of the study region, while the lowest average air temperature, 7.8˚C, was recorded in the western edge of the study area, followed by certain areas in the northwest. One of the reasons for such distribution is that the average air temperature decreases with an increase in altitude, which is consistent with the fact that the terrain in the study area is high in the northwest and low in the southeast. Fig 5B shows the long-term trend of air temperature for each pixel from 2001 to 2020. There were significant geographical differences in air temperature in the study area. From 2001 to 2020, the air temperature showed a warming trend ($k > 0$˚C·a$^{-1}$) in the southeast and northwest of the study area, accounting for 75.8% of the entire region, and these regions passing the 0.05 significance level test. In general, the southeast had the fastest rising air temperature followed by the northwest with the second fastest rising air temperature. The decreasing area of air temperature change rate is in the central and western parts of the study region and presents a regional distribution from northeast to southwest and west. However, the decreasing area of the long-term trend is spatially discontinuous.

Fig 6 shows the interannual variation in the average air temperature from 2001 to 2020. The annual average air temperature was lower in 2012, showing a large change, and the temperature increased after 2012. The five years (5a) moving average eliminated abrupt points and reflected the overall change in temperature. The $R^2$ of linear fitting of annual average temperature and 5-year moving average air temperature are 0.25 and 0.76 respectively, and the growth rates are 0.047˚C/a and 0.055˚C/5a respectively, which shows that the air temperature in the study area is gradually increasing. The significance levels of the annual mean temperature and the 5-year moving average air temperature were both less than 0.05, which passed the significance level test, indicating that the temperature in the study area had a significant increasing trend. The variation trend of air temperature is similar to previous research findings in Tianshui on the interannual variation of annual average air temperature [51]. Existing studies have shown that the global air temperature are rising [52], and the daily maximum and minimum air temperatures in northwest China and the Qinghai-Tibet plateau have risen as well [53]. This study is a good example of this trend.

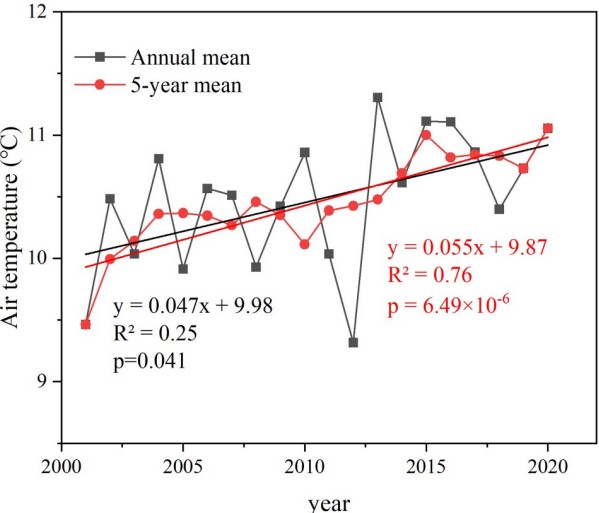

**Fig 6. Interannual variation of average temperature.**

## Seasonal variation of air temperature

There were significant seasonal fluctuations in air temperature changes. Therefore, by analyzing the air temperature change slopes ($k$) in different seasons (Fig 7), we can more accurately show the temporal and spatial change patterns of air temperature. Fig 6 shows that on the spatial scale, the warming trend of air temperature in different seasons according to the order of area occupied. Pixel statistics shows that the areas with positive seasonal variation slopes ($k > 0°C·a^{-1}$) accounted for 92.86% (autumn), 77.51% (summer), 60.74% (spring), and 25.14% (winter) of the total area, respectively. The average air temperature of different seasons was calculated and M-K test was used. The results showed that the air temperature of different seasons had a significant increasing trend in summer and passed the test of significance level of 0.05, while the temperature of spring, autumn and winter had no significant increasing trend and failed the test of significance level of 0.05 (Table 2).

The regions with rising spring air temperatures were mainly distributed in the west, northwest, and north, was advantageous at the start of the crop growth season but exacerbated the drought degree in the spring. The southeast is covered with forests, the air temperature rises in spring, there is less precipitation, and the water content in the vegetation is less, which is more likely to cause forest fires. Therefore, fire monitoring should be strengthened. The air temperature in some areas of Qinan county and Gangu county from northeast to southwest shows a downward trend. This drop in air temperature causes an adverse impact on agriculture vulnerable to low-temperature freezing damage, resulting in serious loss of crops and cash crops, especially frost disasters during the flowering of fruit trees from early April to early May [54], which will have an adverse impact on agricultural production and the development of the local economy. During summer, the air temperature in the western and eastern marginal areas showed a decreasing trend. The eastern margin is the Guanshan Mountains, and the headwater area of the Weihe River is in the west. The altitude is higher than in the central and southeastern areas, indicating that air temperature is affected by altitude. In addition, the air temperature in most other regions is on the rise, possibly because of increase in the constructed area, population growth, and greenhouse gas emissions. Although the intensity of the air temperature rise in autumn is not as noticeable as in summer and spring, the warming area's covering area is the largest. Except for scattered regions in the southeast, west, and

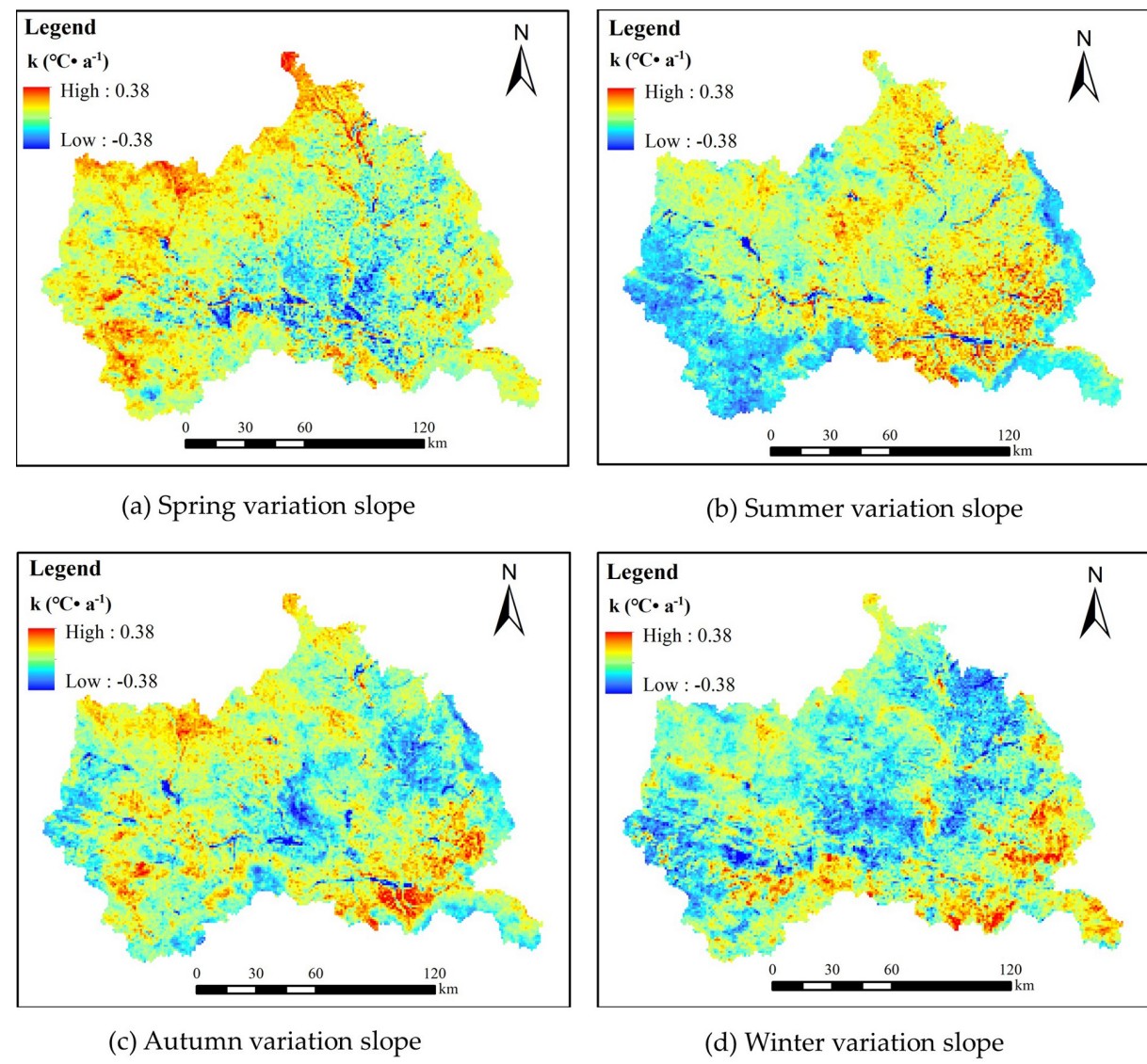

(a) Spring variation slope

(b) Summer variation slope

(c) Autumn variation slope

(d) Winter variation slope

**Fig 7. Seasonal variation slope of Ta from 2001 to 2020.**

northwest, where the slope *k* value was high, the *k* value of most other areas was not very high, indicating that the air temperature rise in autumn was modest. There are many scenic spots in the study area, and autumn is the best time to visit. The emission of automobile exhaust is one of the possible reasons for the temperature rise. The areas primarily affected by winter air temperature increase are mainly located on the southeastern, southern, and northwestern fringes.

**Table 2. Results of the M-K test in different seasons (significant level is 0.05).**

| Season | *p* value | Test value | Trend |
|---|---|---|---|
| spring | 0.92 | 0.097 | No trend |
| summer | 0.0034 | 2.92 | Increasing |
| autumn | 0.086 | 1.71 | No trend |
| winter | 0.63 | -0.48 | No trend |

The reason for the air temperature increase in winter may be related to the weakening of the Siberian high system in this region [55], and the air temperature has a significant negative correlation with the intensity of the Siberian high pressure [56]. The weakening of the intensity of the Siberian high pressure inevitably leads to an increase in air temperature in winter.

## Discussion

### Analysis of influence of NDVI on air temperature estimation

The S-G filtering method not only supplements the missing pixel values and eliminates the outliers in the reconstruction of NDVI time series data but also smoothen the steep rise or drop in values considered as noise, which corrected the original NDVI curve well. The time series curve before and after reconstruction was close to the actual vegetation growth. The long-term trend of NDVI reflects the trend of vegetation cover change, whereas the spatial variability shows the complexity and heterogeneity of the spatial structure change of vegetation cover. According to pixel statistics, the inter-annual change rate, $k$, of NDVI showed an increasing trend from 2001 to 2020, accounting for 98.64% of the entire study area, and only 1.36% of the area showed a decline (Fig 8A). From 2001 to 2020, some regions in the east, southwest, and west of the study area showed small variability, with a minimum value of 0.018 (Fig 8B), indicating that the vegetation cover changes in these areas was not significant. The southeastern region of the study area had the most variety in vegetation cover, with a trend of slowly increasing, obvious increasing, slowly increasing, and obvious increasing from southeast to northwest. Closer to the northwest, the variation in vegetation cover is more obvious, and the maximum variation coefficient of some regions of Longxi and Tongwei reaches 0.37. The Fig 8C shows the change trend of NDVI in the study area. The annual mean value of NDVI in the study area increased significantly in the past 20 years, with an average growth rate of 0.0052/a, which passed the test of significance level of 0.05 (P<0.05) and Z value of 4.74.

The correlation between NDVI and Ta reveals the relationship between them, and the impact of NDVI on air temperature can be analyzed from this perspective. In terms of spatial distribution, the correlation coefficient between NDVI and Ta at the pixel scale was larger in the east than in the west, and larger in the southeast than in the northwest (Fig 9). There was a negative correlation between Ta and NDVI in scattered areas in northwest and northwest China (-0.7< $R$ <0), especially near the source of the Weihe River with higher elevation in west China. However, the NDVI in this region showed a trend of slow increase, and the long-term trend $k$ was between 0 and 0.003 (Fig 8A). This indicates that an increase in NDVI may lead to a decrease in air temperature. The regions that passed the significance level test ($p < 0.05$) were mainly distributed in the east and south of the study area with high correlation coefficient $R$.

There was a positive correlation between NDVI and Ta in most areas of the study area ($R > 0$). Pixel statistics show that areas with $R$>0 account for 94.34% of the total area. The correlation coefficient of $R > 0$ is divided into four grades (0< $R$ ≤0.3, 0.3< $R$ ≤0.5, 0.5< $R$ ≤0.7, 0.7< $R$ ≤1) (Fig 9). The region with the largest correlation coefficient was distributed from Guanshan Mountain to Maiji Mountain in the southeast. The NDVI in this region is significantly high, and the long-term trend, $k$, was between 0.009 and 0.017. An increase in air temperature had a positive effect on vegetation growth in this region. In the river valley, the correlation between NDVI and Ta was strong ($R$ >0.5), but the increase in NDVI was not obvious. This is due to the increased urbanization in the river valley with economic and social growth, resulting in slight decline in vegetation coverage and a decreasing trend in the long-term trend (- 0.01< $k$ <0). The sloping land on both sides of the Weihe River valley has curbed

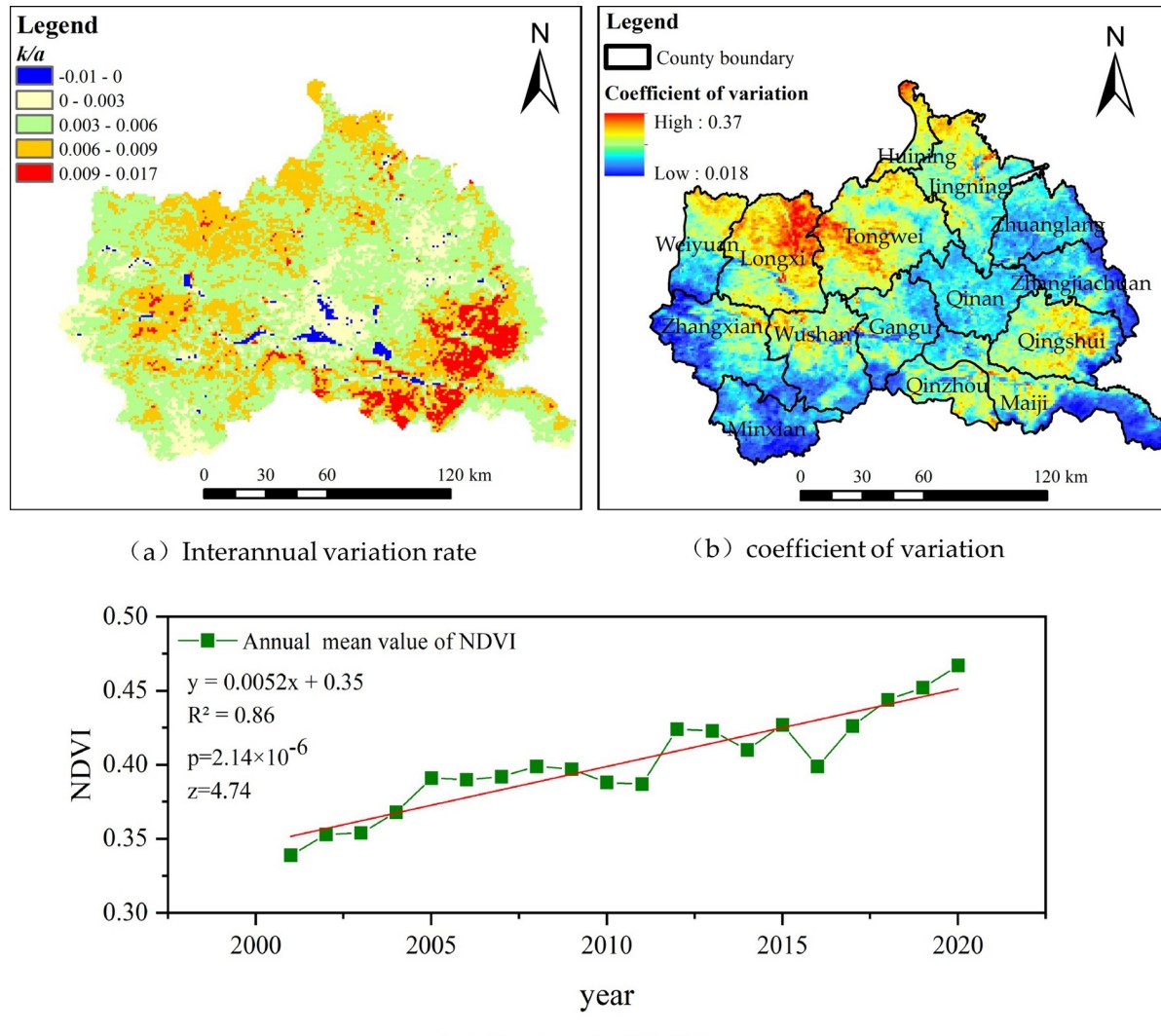

（a）Interannual variation rate                    （b）coefficient of variation

（c）The trend of NDVI

**Fig 8. Interannual variation rate, coefficient of variation and change trend of NDVI.**

the deterioration of the ecological environment and greatly promoted vegetation restoration due to the projects of returning farmland to forest, grassland, and barren mountain afforestation. The long-term trend of NDVI in these areas increased slowly ($0 < k < 0.003$). The main reason for the increase in the air temperature in the valley is its location around towns and villages. Man-made structures have a significant impact on air temperature [57]. During the daytime, when the surface absorbs heat from the sun, the surface temperature rises, increasing the land surface heat flux. The heat flux is transmitted from the ground to the colder air above and thus increases the air temperature. In general, vegetation cover increases with an increase in air temperature, which plays a positive role in promoting vegetation growth.

## Analysis of the influence of LST on air temperature estimation

Fig 10A shows the long-term trend of LST for each pixel from 2001 to 2020. The long-term trend of LST exhibits spatial differences. In the valley area where the Weihe River and its tributaries flow, the LST shows an increasing trend ($k > 0$), whereas in most other areas, the

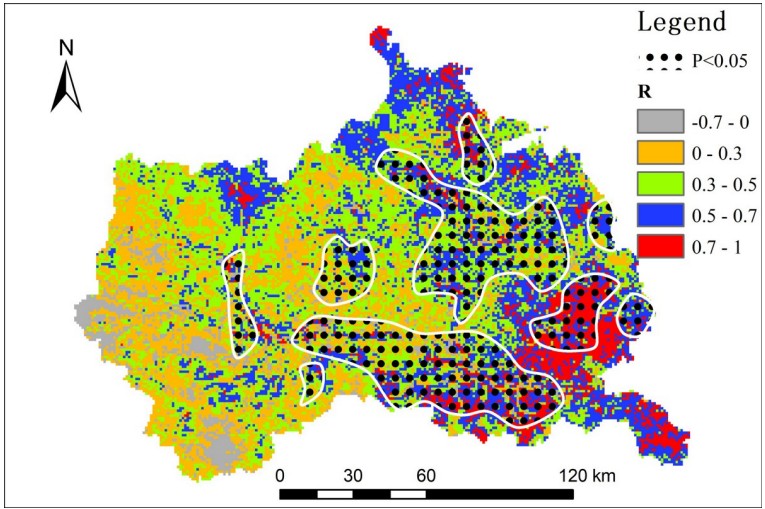

**Fig 9. Correlation coefficient between Ta and NDVI.**

LST shows a decreasing trend ($k < 0$). The significance test results based on M-K show that the significant increase in LST is not apparent and is mainly distributed in the urban area of Tianshui city. A possible reason for this is that the towns and villages are scattered on both sides of the Weihe River, an area with high population. In recent years, with the growth of economy, the construction area of roads, houses and other buildings has increased, and the quality of surface artificial building materials has improved, in terms of its emissivity (which determines the amount of long wave radiation emitted from the earth's surface, and also determines the temperature of the earth's surface), it affects the temperature of the earth's surface, causing it to warm. The areas with a significant LST decrease were mainly distributed in the southeast, south, and west of the study area (Fig 10B), and the areas with no significant LST change were mainly distributed in the middle to northwest of the study area. Pixel statistics shows that the areas with a significant increase, significant decrease, and insignificant change account for 0.01%, 22.91%, and 77.08% of the total area, respectively. The statistics of the $p$ value of the significance test showed that only 45.29% of the pixels passed the test of $p<0.05$ (Fig 10C), which showed that there was no significant change in LST in most regions of the study area. From 2001 to 2020, the interannual LST and NDVI correlation coefficients showed that, in addition to the southeast of Gansu and Shaanxi in the study area, the bordering area correlation coefficient $R$ was positive, and most of the regional correlation coefficient $R$ was less than zero, showing a negative correlation (Fig 10D), which indicates that vegetation cover has a great influence on LST change in the study area. LST decreased, while NDVI increased in the study area, but only part of the southeastern part of the study area passed the significance level test of 0.05. Nevertheless, the correlation between LST and NDVI is consistent with the results of Yu et al. [58], that is, Mann-Kendall significance test and annual mean land surface temperature both show that LST decreases, and it also confirms the trend of wetness in northwest China [59].

Fig 11 shows the correlation between Ta and LST. There was a positive correlation between Ta and LST in most of the study area ($R > 0$). Pixel statistics shows that the area with $R > 0$ accounts for 93.62% of the total area, of which $0< R \leq 0.3$, $0.3 < R \leq 0.5$, $0.5 < R \leq 0.7$ and $R > 0.7$ account for 11.1%, 25%, 38.02% and 19.39% of the total area respectively. The area with $R$ greater than 0.5, which was more than half of the study area, was almost in the area where the interannual change of LST was not significant, which indicates that Ta may be

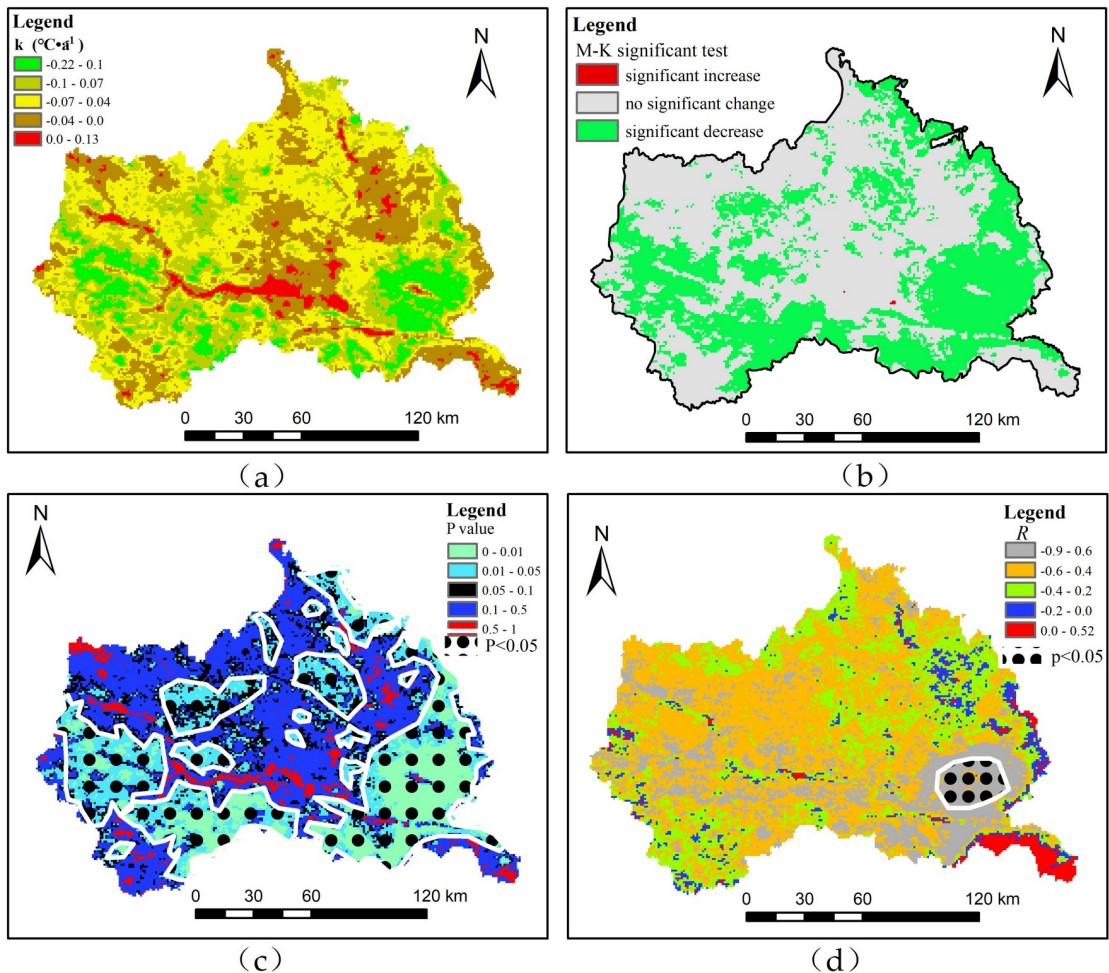

**Fig 10. Long-term trend of LST, M-K change detection, *P* value, correlation coefficient distribution of LST and NDVI from 2001 to 2020.** (a) long-term trend of LST, (b) M-K change detection, (c) *p* value, (d) spatial distribution of correlation coefficient between LST and NDVI.

restricted by other factors in addition to the influence of LST. However, the region with correlation coefficient *R* greater than 0.5 passed the significance level test of 0.05. The southeastern part of the study area is a mountainous and valley landform, with low terrain. Under the influence of solar radiation, more sunlight and heat were collected during the day, and the average temperature was high because of the local circulation effect of the valley wind. However, there was a negative correlation between Ta and LST in the southwest of Guanshan Mountain and Maijishan Mountain in the southeast of the study area ($R < 0$) (Fig 11), indicating variation between LST and Ta. The main reason is that these areas are distributed in forests with high vegetation coverage. Because of the interception of the vegetation canopy, sunlight cannot directly enter the surface, and the ground cannot absorb heat, thus the ground surface temperature remains low. Previous studies have shown that high vegetation coverage plays a role in cooling the surface [60]. The air temperature estimation results show that the air temperature in the southeast increases, which means that the land surface temperature decreases vertically, and not necessarily the air temperature directly above it. This is because the air temperature estimation is the result of the joint action of multiple factors, such as NDVI, albedo, DEM, and the geographical location of the study area.

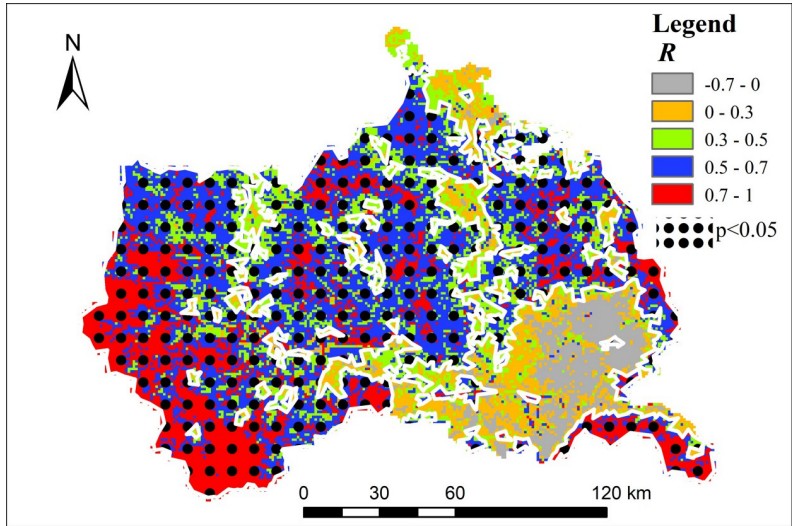

**Fig 11. The correlation coefficient between Ta and LST.**

## Analysis of influence of DEM on air temperature estimation

The rate of temperature change with altitude is the air temperature lapse rate (TLR) [61]. Fig 11A shows the relationship between the DEM and air temperature. It can be seen from the figure that the air temperature decreases by 0.27°C every hundred meters (hm) of DEM increases, that is, the TLR is 0.27°C, and the determination coefficient $R^2$ is 0.96. There is a high correlation between air temperature and elevation, which indicates that air temperature is highly dependent on the DEM. This study result is lower than the annual reference field of land surface temperature variation in mainland China, that is, the air temperature decreases by 0.51°C [62], with an increase of 100 m altitude, the free air TLR is 0.65°C/hm [63], and the near-surface air temperature has a variation value (0.452–0.656°C/hm) [64]. The main reason can be summarized as follows: (1) the mean value of DEM and Ta within every 100 m is calculated, and the air temperature is instantaneous, which is different from the daily average air temperature; (2) the topography of the study area is undulated. From the mountainous region in the southeast, the central valley to the loess hilly and gully region in the northwest, sunlight is affected by the terrain shadow, sloping slope surface, and monsoon, and the sensible heat flux changes are not extreme, while the TLR is slightly small.

The variation of air temperature with elevation can be divided into three sections: DEM < 20.71 hm, 20.71 hm < DEM <25.81 hm and DEM > 25.81 hm (Fig 12A). When DEM < 20.71 hm and DEM>25.81 hm, the slope of the linear relationship between air temperature and DEM is higher, the TLR are 0.37°C and 0.38°C respectively, higher than 0.27°C (Fig 12B and 12D), and the determination coefficient $R^2$ is 0.95, indicating that the change in air temperature in these two elevation zones is obvious. When 20.71 hm < DEM < 25.81 hm, the relationship between air temperature and DEM does not change significantly. For every 100 m increase, the air temperature decreases by 0.12°C (Fig 12C). The air temperature drop rate is different in different DEM zoning, mainly because the air temperature is influenced by various factors such as time, geographical location, terrain, precipitation, wind speed, evapotranspiration, and the underlying surface [65, 66]. When DEM < 20.71 hm, on the one hand, the surface receives more solar radiation; on the other hand, the urban construction land is mostly distributed in the low-altitude area, which is affected by various human economic and construction activities on the environment. The near-surface temperature is higher, and the

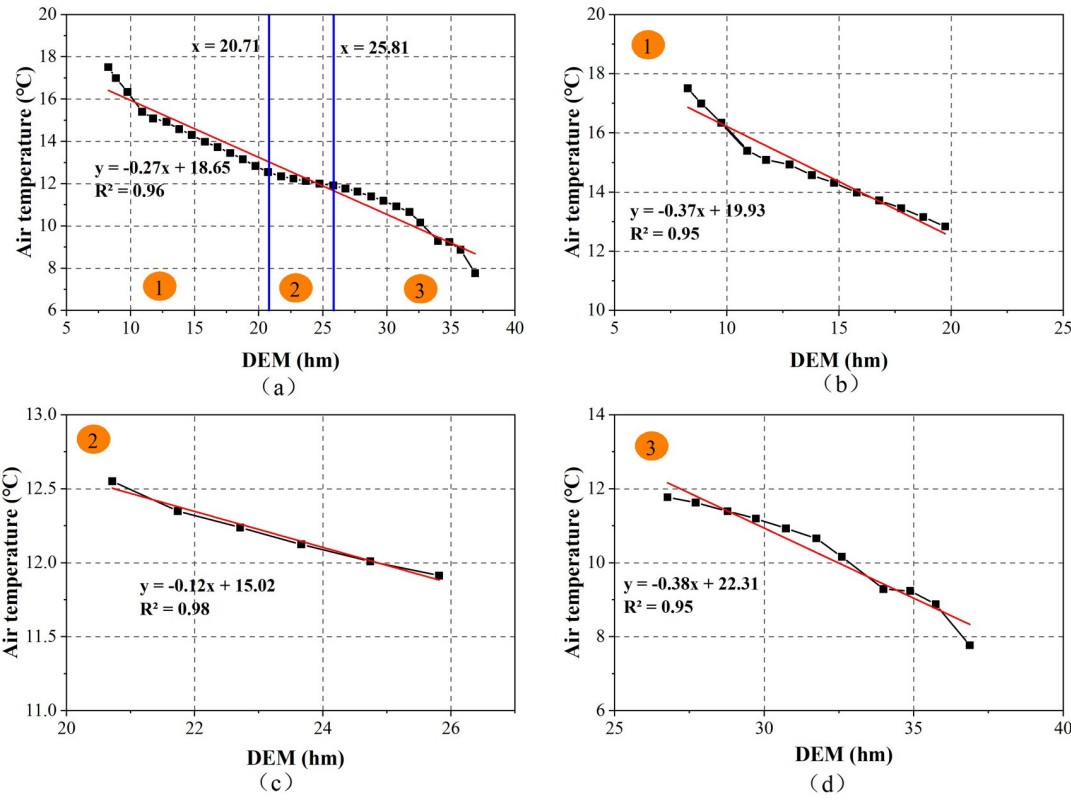

**Fig 12. Comparison diagram of air temperature and DEM.**

TLR changes greatly. The area with 20.71 hm < DEM < 25.81 hm was located in the west and northwest of the study area. The air at higher altitudes may be heated by the latent heat release related to water vapor condensation; thus, the TLR decreases. When DEM > 25.81 hm, with the increase in DEM, there are few human activities, sparse vegetation, high wind speed, rapid change in surface heat, large decline in air temperature, and increase in TLR.

## Uncertainty of air temperature estimation

The estimated temperature of Qingshui meteorological station in the study area is higher than the observed value (Fig 3C), and the RMSE between them is 3.02°C, indicating that the estimated temperature of this station may be inaccurate. The main reason for this condition is, on the one hand, the errors of the satellite images themselves [67] while on the other hand, the air temperature observed at a single station cannot reliably represent the air temperature at different distances due to changes in surface coverage and altitude, and the data itself inevitably has errors; therefore, removing outliers from the data is worth considering. Despite this, the RMSE and MAE values of other sites in the study area are relatively small (Fig 3A, 3B and 3D), which means that the near-surface air temperature estimated by the BP-ANN method has high estimation accuracy and produces accurate results. In previous studies, remote sensing inversion of mean or maximum air temperature was carried out on time scales of on time scales of daily, monthly, and yearly [68], and the inversion accuracy was higher than instantaneous air temperature. For example, Kilibarda et al. [69] interpolated the temporal and spatial values of daily mean air temperature in global land areas including Antarctica, and showed that RMSE of daily mean air temperature with spatial resolution of 1 km was 2.4°C, which was slightly

higher than RMSE of 2.28˚C and 1.86˚C of Wushan and Weiyuan stations in this study area. This is attributed to the instantaneous time air temperature estimation of satellite transit at regional scale studied in this paper.

## Conclusion

Near-surface air temperature is one of the most important parameters in climate, ecological, and hydrological research with agricultural applications. The estimation of air temperature at appropriate spatiotemporal scales is an effective supplement to the air temperature measurement results of conventional stations. This study used interpolated and reconstructed LST and NDVI data and other remote sensing data to estimate the instantaneous air temperature with a time scale of 8 days based on the BP-ANN algorithm and multiple regression method, which is of great importance for the study of regional environmental monitoring. Through the estimation of the air temperature on a regional scale, the following conclusions were obtained:

1. A comparison between the air temperature estimation results and the observed values of the stations showed that the accuracy of the air temperature estimation was high, and the correlation coefficient $R$ was greater than 0.97. Except for the RMSE of the Qingshui meteorological station being slightly greater than 3˚C, the RMSE of the other stations was less than 2.5˚C, and the MAE was less than RMSE and less than 2˚C, which indicated that the estimated near-surface temperature results were accurate.

2. The temperature of the air in space was increasing but affected by the NDVI, LST, and DEM. NDVI had a positive effect on the increase in air temperature. There was a positive correlation between LST and Ta, but for the area covered by surface forest, LST and Ta showed Earth's atmosphere differences. The air temperature decreased with an increase in the DEM, but the terrain of the study area was extremely complex. The TLR was lower than 0.65˚C/hm of free air. The effect of the DEM on air temperature was zonal, and the TLR was not consistent in different elevation zones.

In this study, the input parameters of air temperature estimation included LST, NDVI, albedo, and DEM. However, air temperature estimation is a complex task and it is very likely that more input parameters will be needed for the air temperature estimation model in the future to improve the accuracy of air temperature estimation.

## Author Contributions

**Data curation:** Liu Baokang.

**Formal analysis:** You Xiaoni.

**Methodology:** Shi Yaya.

**Software:** Wang Runke.

**Validation:** Wu Chengyong.

**Writing – original draft:** Wang Runke.

**Writing – review & editing:** Wang Runke.

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
