## [Decision Letter · Decision Letter 0]

1 May 2022

PONE-D-22-09843Study on air temperature estimation and its influencing factors in a complex mountainous areaPLOS ONE

Dear Dr. Wang,

Thank you for submitting your manuscript to PLOS ONE. After careful consideration, we feel that it has merit but does not fully meet PLOS ONE’s publication criteria as it currently stands. Therefore, we invite you to submit a revised version of the manuscript that addresses the points raised during the review process.

We look forward to receiving your revised manuscript.

Kind regards,

Ming Luo, Ph.D

Academic Editor

PLOS ONE

Journal Requirements:

4. Please upload a copy of Figures 1 to 11, to which you refer in your text. If the figure is no longer to be included as part of the submission please remove all reference to it within the text.

Reviewers' comments:

Reviewer's Responses to Questions

**Comments to the Author**

1. Is the manuscript technically sound, and do the data support the conclusions?

Reviewer #1: Yes

Reviewer #2: Partly

2. Has the statistical analysis been performed appropriately and rigorously? 

Reviewer #1: Yes

Reviewer #2: Yes

3. Have the authors made all data underlying the findings in their manuscript fully available?

Reviewer #1: Yes

Reviewer #2: No

4. Is the manuscript presented in an intelligible fashion and written in standard English?

Reviewer #1: Yes

Reviewer #2: No

5. Review Comments to the Author

Reviewer #1: The manuscript by Runke Wang et al., attempt to estimate the near-surface air temperature (Ta) in a complex mountainous area of China by using NDVI, LST, surface albedo, and DEM as the input data and the BP-ANN method. The authors created the Ta dataset with high spatial resolution and investigated its spatial and temporal changes. Moreover, the possible effects of NDVI, LST, and DEM on Ta were also discussed. These findings are of great importance for the regional environmental monitoring in the mountainous area. I would like to recommend an acceptation after these comments as follows are addressed.

Major comments:

(1) In the Abstract, Lines 28-33: The authors discussed the possible effects of NDVI, LST, and DEM on the air temperature estimation. However, the descriptions over here are not good as the possible mechanisms of the influences of these factors on air temperature are not clearly clarified. Thus, I suggest the authors to rewrite this part to improve the distributions.

(2) In the Materials and Methods, Lines 176-179: A total of 14 meteorological stations are used in the study area and the number and the number of stations is small. In this study, the authors divided the 14 stations into two categories. Ten of them were used to drive the model for estimating the near-surface temperature, and the four remaining stations were used to estimate the performance of the model. As the number of the total meteorological stations is small, the above method is not perfect. Thus, I suggest the authors to used ten-fold cross validation method to build the model and then to estimate the performance model. Ten-fold cross validation would perform the fitting procedure a total of ten times, with each fit being performed on a training set consisting of 90% of the total training set selected at random, with the remaining 10% used as a hold out set for validation. If the authors have done the research by ten-fold cross validation method, they can exhibit the new results in comparison with the original results that were obtained based on the previous method.

(3) In the results and analysis, it is necessary to perform significance tests for trend analysis as well as correlation analysis. For instance, (1) the significant level of Figure 4b should be given, in which you can exhibit the trend passing 0.05 significant level by dots. In addition, the unit of Figure 4b is not correct. Please check it. In my opinion, "interannual change rate" should be "long-term trend", please check it. (2) In Figure 5, it is necessary to give the p-values for the linear trends as we should know the significant level when we assessing the long-term change. (3) In Figure 6, please give the significant level of long-term trends in air temperatures during four seasons. In addition, please express the related contents in the full manuscript according to atmospheric terminology. (4) In Figure 7, please give the significant level of long-term trend of NDVI. (5) In Figure 8, please give the significant level of correlation coefficient between Ta and NDVI and the region passing the significant test at 0.05 level should be marked by dots. (6) In Figures 9 and 10, please give the significant level and the region passing the significant test at 0.05 level should be marked by dots.

Specific comments:

(1) The "study area" is found in many places in the abstract, while I cannot find any description of the "study area". It is necessary to give specific information about the study area in the abstract.

(2) Line 148: divided -> defined

(3) Line 152: Data and processing -> Data

(4) Line 153: Remote sensing data and preprocessing -> Remote sensing data

(5) Line 160: land cover/land cover -> land use/land cover

(6) Line 169: please check the "re obtained"

(7) Line 170: Ground data and preprocessing -> Ground data

(8) Line 183: Air temperature estimation method -> Air temperature estimation

(9) Lines 258-259: The larger the CV, the more obvious the change in NDVI. Please check this sentence carefully.

(10) Lines 263-264: There are many grammatical errors in the sentence. Please check it carefully and rewrite this sentence.

(11) Line 280: Results and analysis -> Results

(12) Lines 282-283: 9580 meteorological stations were selected to establish the equation, while in the data (Figure 1) the number of total meteorological stations is 14. Please it carefully.

(13) Line 528: Uncertainty of air temperature estimation results -> Uncertainty of air temperature estimation.

(14) Lines 529-531: what does it mean that the RMSE value of all estimated air temperature at the regional scale was still large than the ground observation value? Please check the sentence carefully.

(16) Line 538: on time scales of day, month, and year -> on time scales of daily, monthly, and yearly

(17) Lines 539-541: For example, studies have shown that ... Please check the sentence carefully.

Reviewer #2: The idea and practice of applying (1) remote sensing data like MODIS LST, emissivity, and NDVI, (2) gridded products like land cover and land use, and dem, (3) observation data to build near surface air temperature dataset through the BP-ANN algorithm in Tianshui during 2001~2020. This work lacks scientific merit and the manuscript does not make any significant improvement to the intended field of research.

6. PLOS authors have the option to publish the peer review history of their article (what does this mean?). If published, this will include your full peer review and any attached files.

Reviewer #1: No

Reviewer #2: No

---

## [Author Response · Author response to Decision Letter 0]

30 May 2022

Reviewer #1: The manuscript by Runke Wang et al., attempt to estimate the near-surface air temperature (Ta) in a complex mountainous area of China by using NDVI, LST, surface albedo, and DEM as the input data and the BP-ANN method. The authors created the Ta dataset with high spatial resolution and investigated its spatial and temporal changes. Moreover, the possible effects of NDVI, LST, and DEM on Ta were also discussed. These findings are of great importance for the regional environmental monitoring in the mountainous area. I would like to recommend an acceptation after these comments as follows are addressed.

Major comments:

(1) In the Abstract, Lines 28-33: The authors discussed the possible effects of NDVI, LST, and DEM on the air temperature estimation. However, the descriptions over here are not good as the possible mechanisms of the influences of these factors on air temperature are not clearly clarified. Thus, I suggest the authors to rewrite this part to improve the distributions.

(2) In the Materials and Methods, Lines 176-179: A total of 14 meteorological stations are used in the study area and the number and the number of stations is small. In this study, the authors divided the 14 stations into two categories. Ten of them were used to drive the model for estimating the near-surface temperature, and the four remaining stations were used to estimate the performance of the model. As the number of the total meteorological stations is small, the above method is not perfect. Thus, I suggest the authors to used ten-fold cross validation method to build the model and then to estimate the performance model. Ten-fold cross validation would perform the fitting procedure a total of ten times, with each fit being performed on a training set consisting of 90% of the total training set selected at random, with the remaining 10% used as a hold out set for validation. If the authors have done the research by ten-fold cross validation method, they can exhibit the new results in comparison with the original results that were obtained based on the previous method.

(3) In the results and analysis, it is necessary to perform significance tests for trend analysis as well as correlation analysis. For instance, (1) the significant level of Figure 4b should be given, in which you can exhibit the trend passing 0.05 significant level by dots. In addition, the unit of Figure 4b is not correct. Please check it. In my opinion, "interannual change rate" should be "long-term trend", please check it. (2) In Figure 5, it is necessary to give the p-values for the linear trends as we should know the significant level when we assessing the long-term change. (3) In Figure 6, please give the significant level of long-term trends in air temperatures during four seasons. In addition, please express the related contents in the full manuscript according to atmospheric terminology. (4) In Figure 7, please give the significant level of long-term trend of NDVI. (5) In Figure 8, please give the significant level of correlation coefficient between Ta and NDVI and the region passing the significant test at 0.05 level should be marked by dots. (6) In Figures 9 and 10, please give the significant level and the region passing the significant test at 0.05 level should be marked by dots.

Specific comments:

(1) The "study area" is found in many places in the abstract, while I cannot find any description of the "study area". It is necessary to give specific information about the study area in the abstract.

(2) Line 148: divided -> defined

(3) Line 152: Data and processing -> Data

(4) Line 153: Remote sensing data and preprocessing -> Remote sensing data

(5) Line 160: land cover/land cover -> land use/land cover

(6) Line 169: please check the "re obtained"

(7) Line 170: Ground data and preprocessing -> Ground data

(8) Line 183: Air temperature estimation method -> Air temperature estimation

(9) Lines 258-259: The larger the CV, the more obvious the change in NDVI. Please check this sentence carefully.

(10) Lines 263-264: There are many grammatical errors in the sentence. Please check it carefully and rewrite this sentence.

(11) Line 280: Results and analysis -> Results

(12) Lines 282-283: 9580 meteorological stations were selected to establish the equation, while in the data (Figure 1) the number of total meteorological stations is 14. Please it carefully.

(13) Line 528: Uncertainty of air temperature estimation results -> Uncertainty of air temperature estimation.

(14) Lines 529-531: what does it mean that the RMSE value of all estimated air temperature at the regional scale was still large than the ground observation value? Please check the sentence carefully.

(16) Line 538: on time scales of day, month, and year -> on time scales of daily, monthly, and yearly

(17) Lines 539-541: For example, studies have shown that ... Please check the sentence carefully.

Reviewer #2: The idea and practice of applying (1) remote sensing data like MODIS LST, emissivity, and NDVI, (2) gridded products like land cover and land use, and dem, (3) observation data to build near surface air temperature dataset through the BP-ANN algorithm in Tianshui during 2001~2020. This work lacks scientific merit and the manuscript does not make any significant improvement to the intended field of research.

---

## [Editor Report · Decision Letter 1]

30 Jun 2022

PONE-D-22-09843R1Study on air temperature estimation and its influencing factors in a complex mountainous areaPLOS ONE

Dear Dr. Wang,

Thank you for submitting your manuscript to PLOS ONE. After careful consideration, we feel that it has merit but does not fully meet PLOS ONE’s publication criteria as it currently stands. Therefore, we invite you to submit a revised version of the manuscript that addresses the points raised during the review process.

We look forward to receiving your revised manuscript.

Kind regards,

Ming Luo, Ph.D

Academic Editor

PLOS ONE
---

## [Author Response · Author response to Decision Letter 1]

3 Jul 2022

Reply: Thank you very much

We have already uploaded the figure files to the Preflight Analysis and Conversion Engine (PACE) digital diagnostic tool, and all the figures have passed the inspection and meet PLOS requirements.

---

## [Editor Report · Decision Letter 2]

1 Aug 2022

Study on air temperature estimation and its influencing factors in a complex mountainous area

PONE-D-22-09843R2

Dear Dr. Wang,

We’re pleased to inform you that your manuscript has been judged scientifically suitable for publication and will be formally accepted for publication once it meets all outstanding technical requirements.

Kind regards,

Ming Luo, Ph.D

Academic Editor

PLOS ONE
---

## [Editor Report · Acceptance letter]

5 Aug 2022

PONE-D-22-09843R2 

Study on air temperature estimation and its influencing factors in a complex mountainous area 

Dear Dr. Wang:

I'm pleased to inform you that your manuscript has been deemed suitable for publication in PLOS ONE. Congratulations! Your manuscript is now with our production department. 

Kind regards, 

on behalf of

Dr. Ming Luo 

Academic Editor

PLOS ONE